# Multiple Fingerprints and Spectrum-Effect Relationship of Polysaccharides from *Saposhnikoviae Radix*

**DOI:** 10.3390/molecules27165278

**Published:** 2022-08-18

**Authors:** Mengqi Yu, Guang Xu, Ming Qin, Yanling Li, Yuying Guo, Qun Ma

**Affiliations:** Department of Pharmacy of Chinese Materia Medica, Beijing University of Chinese Medicine, Beijing 102488, China

**Keywords:** *Saposhnikoviae Radix* polysaccharide, multiple fingerprints, quality control, anti-allergic activity, spectrum-effect relationship

## Abstract

**Highlights:**

The multiple fingerprints of 10 batches of *Saposhnikoviae Radix* polysaccharide(SP) were prepared and chemometrics analysis was carried out.The anti-allergic activity of the SPs was evaluated.The spectrum-effect relationship of SPs was establishedd.

**Abstract:**

PMP-HPLC, FT-IR, and HPSEC fingerprints of 10 batches of polysaccharides from *Saposhnikoviae Radix* with different production areas and harvest times have been prepared, and the chemometrics analysis was performed. The anti-allergic activity of 10 batches of *Saposhnikoviae Radix* polysaccharide (SP) was evaluated, and the spectrum-effect relationship of the 10 batches of SP was analyzed by gray correlation degree with the chromatographic fingerprint as the independent variable. The results showed that the PMP-HPLC, HPSEC, and FT-IR fingerprints of 10 batches of SP had a high similarity. Two monosaccharides (rhamnose and galactose), the polysaccharide fragment Mn = 8.67 × 10^6^~9.56 × 10^6^ Da, and the FT-IR absorption peak of 892 cm^−1^ can be used as the quality control markers of SPs. All 10 batches of SP could significantly inhibit the release of β-HEX in RBL-231 cells, and the polysaccharides harvested from Inner Mongolia in the winter had the best anti-allergic activity. The spectrum-effect relationship model showed that the monosaccharide composition and molecular weight were related to the anti-allergic activity of the SPs. Multiple fingerprints combined with spectrum-effect relationship analysis can evaluate and control the quality of SPs from the aspects of overall quality and efficacy, which has more application value.

## 1. Introduction

*Saposhnikoviae Radix* is a traditional Chinese medicine, also called Fangfeng, the dry root of *Saposhnikovia divaricatee* (*Turcz.*) *Schischk*, and is widely used in the clinical treatment of cold, headache, and skin pruritus [1]. *Saposhnikoviae Radix* has a variety of pharmacological activities, such as anti-inflammatory, bacteriostatic, anti-allergy, and anti-oxidation [2,3,4,5]. *Saposhnikoviae Radix* contains a variety of active ingredients, in which coumarins and chromogens have been studied extensively and work has focused on the structure and related activity of these compounds. As one of the main active components, *Saposhnikoviae Radix* polysaccharide (SP) has demonstrated a range of biological activities [6,7,8,9]. However, there are few reports on monosaccharide composition, molecular weight distribution, and structural characteristics of the *Saposhnikoviae Radix* polysaccharide, but there is no research on the quality control of *Saposhnikoviae Radix* polysaccharide [8,10]. At the same time, Polysaccharide quality control is a great challenge due to its complex structure, large molecular weight, and no UV absorption [11,12]. It is significant to establish a method to evaluate the quality of the *Saposhnikoviae Radix* polysaccharide from several aspects such as monosaccharide composition, molecular weight, and molecular weight distribution, characteristic structure, and so on.

In recent years, fingerprint with emphasis on global and fuzzy information analysis has proved to be a convenient and effective way to standardize and control the quality of herbal materials containing complex natural ingredients [13]. The common polysaccharide fingerprint contains the composition of monosaccharides, and the commonly used detection methods include PMP-HPLC, GC-MS, and HPAEC-PAD, among which PMP-HPLC is simple and fast, and most widely used [14]. In addition, there are molecular weight and molecular weight distribution fingerprints and structure characteristics fingerprints, commonly used detection methods are gel permeation chromatography and infrared spectroscopy, respectively [15,16]. Compared to a single fingerprint, multiple fingerprints can provide more comprehensive information on polysaccharides on the types and proportions of monosaccharides, molecular weight distribution, and chemical bond structure, etc. Combined with the chemometrics analysis, the quality control of polysaccharides can be effectively realized. Li et al. established three reference fingerprints of HPSEC, PCD-HPLC, and FR-IR, and combined them with chemometrics to explore the main indicators of quality control to evaluate *Sarcandra glabra* polysaccharide [17]. Li et al. established the multiple fingerprints of 10 batches of *Astragalus* polysaccharides including UPLC/Q-TOF-MS, NMR, and FT-IR fingerprints, combined with the chemometrics analysis, the method, and indexes for quality control of astragalus polysaccharide were obtained [18].

The spectrum-effect relationship is the advanced stage of fingerprint of traditional Chinese medicine [19]. By correlating the polysaccharide fingerprint with its pharmacodynamic ability through mathematical statistics, the main active parts of polysaccharides can be mined and a more reasonable quality control method can be established. Zhang et al. established the HPGPC-ELSD fingerprint and investigated the anti-inflammatory activity of *Lycium barbarum* polysaccharide, and the main anti-inflammatory active parts of *Lycium barbarum* polysaccharide were mined out through spectrum-effect relationship established by the grey correlation analysis method [20].

This study aimed to develop multiple fingerprints of *Saposhnikoviae Radix* polysaccharides consisting of PMP-HPLC fingerprint, HPSEC fingerprint, and FT-IR fingerprint, and screen out the herbal markers combined with chemometrics to establish an effective quality control method. At the same time, the anti-allergic activity of *Saposhnikoviae Radix* polysaccharide was determined, and the spectrum-effect relationship was established to screen out the main anti-allergic parts of *Saposhnikoviae Radix* polysaccharide. There are two innovations in this paper, one is to use multiple fingerprints to achieve comprehensive quality control of *Saposhnikoviae Radix* polysaccharide, and the other one is to construct the spectrum-effect relationship of anti-allergic activity and screen out the main anti-allergic active parts of *Saposhnikoviae Radix* polysaccharide.

## 2. Results and Discussion

### 2.1. PMP-HPLC Fingerprint of SPs and Chemometric Analysis

#### 2.1.1. PMP-HPLC Fingerprint and Similarity Analysis

PMP-HPLC spectra of 10 batches of SP were imported into the similarity evaluation system of TCM chromatographic fingerprints (2012 version). S1 was selected as the reference fingerprint, and the width of the time window was set as 0.1. Mark peak matching was performed by manual multi-point correction, and the standard reference PMP-HPLC fingerprint was generated by the average method. We can see that 10 batches of polysaccharide samples have similar characteristic peak retention times as shown in Figure 1A. At the same time, the characteristic peak area of the 10 batches of SP was different, which indicated that although the 10 batches of SP had the same monosaccharide composition, the monosaccharide proportion was different. Figure 1B showed the standard reference PMP-HPLC fingerprint of SPs with six common characteristic peaks, the first to sixth peaks are the monosaccharides of mannose, rhamnose, galacturonic acid, glucose, galactose, and arabinose, respectively. During the experiment, 0.05 mol L^−1^ phosphoric acid buffer and acetonitrile were used to elute the mixed sample. It was found that the glucose and rhamnose peaks could not be separated. After adjusting the phosphoric acid buffer concentration to 0.1 mol L^−1^, all monosaccharide peaks were clearly separated.

The similarity of 10 batches of SP was calculated by the correlation coefficient (R) method and cosine (cos θ) method, and the results are shown in Appendix A. The similarity coefficient and cosine value of 10 batches of SP were respectively greater than 0.953 and 0.983, which indicated that the 10 batches of SP had a high degree of similarity, and the generated standard reference fingerprint could reflect the basic characteristics of SPs in monosaccharide composition. According to the similarity analysis of the monosaccharide composition parameters of different SP samples, 10 batches of SP had similar monosaccharide composition.

Different polysaccharides have different monosaccharide compositions, and the detection of the monosaccharide composition is an important step to control the quality of polysaccharides. HPLC method is one of the common, simple, and quick methods to analyze the monosaccharide composition of polysaccharides obtained from nature [21]. HPLC fingerprint of polysaccharides with derivatization has been considered an important part of the identification and quality control of polysaccharides. Our experimental results showed that six monosaccharides, including mannose, rhamnose, galacturonic acid, glucose, galactose, and arabinose were detected in SPs. In the early stage, our group used UPLC-MS/MS to detect the SPs, except for the six monosaccharides mentioned above, trace amounts of ribose and fucose were also detected [22]. Compared with UPLC-MS/MS method, the PMP-HPLC method sacrificed part of sensitivity, but it was easier and quicker for the quality control of SPs.

#### 2.1.2. Chemometric Analysis of PMP-HPLC Fingerprint

Taking the area of the common peaks in the PMP-HPLC fingerprint of SPs as the variable, the Ward method was used for clustering, and the square Euclidean distance method was used as the classification basis. When the discriminant distance was 10, there were two kinds of polysaccharide clusters. Among them, S1, S2, S3, S4, and S5 are the first class, and S6, S7, S8, S9, and S10 are the second class, as shown in Figure 2A. The samples from Heilongjiang and Jilin were basically grouped together, but Inner Mongolia appears in both the first class and second class. The SPs from the same origin can be clustered and there was also a cross-producing area, which indicated that there was a certain but not absolute correlation between the fingerprint of SPs and their producing areas from the perspective of the peak area of monosaccharide composition.

Taking the peak area of PMP-HPLC spectra of 10 batches of SPs as the variable, the principal component analysis showed that the contribution rate of the first two principal components (PC 1 and PC 2) was 80.677% (>70%), so the PC 1 and PC 2 were selected for evaluation. The principal component load matrix reflected the contribution of each variable to the principal component and the direction, the synergistic action of multiple components leads to the quality difference of SPs. All six variables were correlated with PC 1 and PC 2, among which common peaks 1, 4, and 5 contributed more to PC 1, common peaks 2 and 3 contributed greatly to PC 2, and common peaks 1, 4, and 6 were negatively correlated with PC 2, as shown in Appendix A. The PCA score showed that the 10 batches of SP were divided into two groups, S1, S2, S3, S4, and S5 are the one group, and S6, S7, S8, S9, and S10 are the other group. The results were consistent with the HCA analysis, as shown in Figure 2B,C.

In order to find the difference markers between different batches of SP, the PLS-DA model was used. The results showed that R^2^X (cum) = 0.795, R^2^Y (cum) = 0.977, and Q^2^ (cum) = 0.893, all greater than 0.5, indicating that the model was stable and reliable. the PLS-DA scores showed that the 10 batches of SPs were divided into two groups, S1, S2, S3, S4, and S5 are the one group, and S6, S7, S8, S9, and S10 are the other group. The results were consistent with the HCA analysis and PCA analysis, as shown in Figure 2D,E. VIP value was used as the screening criteria to obtain the landmark monosaccharide ingredients that differed between different places of origin. As can be seen from Figure 2F, the VIP values of peak 2 and peak 5 are 1.535 and 1.366 (both greater than 1.0), which are rhamnose and galactose, respectively. It is speculated that these two monosaccharides have a greater impact on the quality of the sample.

### 2.2. HPSEC Fingerprint of SPs and Chemometric Analysis

#### 2.2.1. HPSEC Fingerprint and Similarity Analysis

By establishing the HPSEC fingerprint, we were able to compare the molecular weight distribution differences of SPs in different regions. HPSEC spectra of 10 batches of SP were imported into the similarity evaluation system of TCM chromatographic fingerprints (2012 version), and the standard reference HPSEC fingerprint was generated by the median method. Ten batches of SP have similar characteristic peak retention times as shown in Figure 3A. From the standard reference HPSEC fingerprint of SPs (Figure 3B), SP samples had a high molecular weight main peak, a high molecular weight small peak, and a low molecular weight small peak. According to the molecular weight correction curve fitted by the GPC method, the Mn of the high molecular weight main peak were about ranging from 8.67 × 10^6^~9.56 × 10^6^ Da, and the two small peaks were ranging from 2.50 × 10^6^~3.15 × 10^6^ Da and 1.69 × 10^3^~4.00 × 10^3^ Da respectively. The Mw of the high molecular weight main peak were about ranging from 8.91 × 10^6^~9.74 × 10^6^ Da, and the two small peaks were ranging from 3.02 × 10^6^~3.50 × 10^6^ Da and 8.09 × 10^3^~1.50 × 10^4^ Da respectively. Meanwhile, the average peak area of the main peak is 86.15%, which is much higher than that of the two small peaks, the molecular weight and molecular weight distribution of SP in 10 batches are shown in Appendix A. Therefore, the main peak with high molecular weight could be considered the main active component in SPs.

The similarity of 10 batches of SP was calculated by the correlation coefficient (R) method and cosine (cos θ) method, and the results are shown in Appendix A. The similarity coefficient and cosine value of 10 batches of SP were respectively greater than 0.998 and 0.999, which indicated that the 10 batches of SP had a high degree of similarity, and the generated standard reference fingerprint could reflect the basic characteristics of SPs in molecular weight distribution. According to the similarity analysis of the HPSEC fingerprints, the molecular weight distribution of different SP samples is very similar.

Molecular weight distribution and structure are the important basic information of polysaccharides and are also an indispensable link in the identification and quality control of polysaccharides. Li et al. prepared the HPSEC fingerprint and FT-IR fingerprint of polysaccharides extracted from Zishen Yutai Pills and combined them with the HPLC fingerprint to control the quality of polysaccharide components. The results showed that the method was stable and feasible [23].

#### 2.2.2. Chemometric Analysis of HPSEC Fingerprints

Taking the area of the common peaks in the HPSEC fingerprint of SPs as the variable, the Ward method was used for clustering, and the square Euclidean distance method was used as the classification basis. When the discriminant distance was 10, there were three kinds of polysaccharide clusters. Among them, S6, S8, and S10 are the first class, S2, S4, S5, S7, and S9 are the second class, and S1 and S3 are the third class, as shown in Figure 4A. The SPs from Heilongjiang, Jilin, and Inner Mongolia cannot be grouped according to the origin, which indicates that there was little correlation between the molecular weight distribution of SPs and their producing areas.

Taking the peak area of HPSEC spectra of 10 batches of SP as the variable, the principal component analysis showed that the contribution rate of the first two principal components (PC 1 and PC 2) was 76.647% (>70%), so the PC 1 and PC 2 were selected for evaluation. All three variables were correlated with PC 1 and PC 2, among which common peaks 1 and 3 contributed more to PC 1, common peak 2 contributed greatly to PC 2, and common peak 2 is negatively correlated with PC 1, as shown in Appendix A. The PCA score showed that the 10 batches of SP were divided into three groups, S6, S8, and S10 are the one group, S2, S4, S5, S7, and S9 are the one group, and S1 and S3 are another group. The results were consistent with the cluster analysis, as shown in Figure 4B,C.

To find the difference markers between different batches of SP, the PLS-DA model was used. The results showed that R^2^X (cum) = 0.737, R^2^Y (cum) = 0.736, and Q^2^ (cum) = 0.408, the scores were shown in Figure 4D,E. The PLS-DA scores showed that the 10 batches of SP were divided into three groups, S6, S8, and S10 are the one group, S2, S4, S5, S7, and S9 are the one group, and S1 and S3 are another group. The results were consistent with the HCA analysis and PCA analysis. VIP value was used as the screening criteria to obtain the landmark molecular weight range that differed between different places of origin. As can be seen from Figure 4F, the VIP values of peak 1 and peak 2 are 1.156 and 1.091 (both greater than 1.0), which numbers mean Mn of the two peaks ranging from 8.67 × 10^6^~9.56 × 10^6^ Da and 2.50 × 10^6^~3.15 × 10^6^ Da, respectively.

The results of cluster analysis and principal component analysis showed that there were differences in molecular weight distribution among different SP samples. Two different polysaccharide fragments were screened by PLS-DA analysis, which were Mn = 8.67 × 10^6^~9.56 × 10^6^ Da and Mn = 2.50 × 10^6^~3.14 × 10^6^ Da.

### 2.3. FT-IR Fingerprint of SPs and Chemometric Analysis

#### 2.3.1. FT-IR Fingerprint and Similarity Analysis

FT-IR data of 10 batches of SP were imported into Origin 9.0 software to generate FT-IR fingerprint and standard reference FT-IR fingerprint, as shown in Figure 5. Figure 5A showed that the infrared spectrum characteristics of 10 batches of SP were similar, but there were still some differences in the fingerprint region of 1300–650 cm^−1^. According to the standard reference FT-IR fingerprint (Figure 5B), there were 10 common characteristic peaks in 10 batches of SPs, and the wave number of the characteristic peaks were 3389 cm^−1^, 2935 cm^−1^, 1743 cm^−1^, 1621 cm^−1^, 1423 cm^−1^, 1374 cm^−1^, 1238 cm^−1^, 1079 cm^−^^1^, 1024 cm^−1,^ and 892 cm^−1^ respectively. Among them, the absorption peak at 3389 cm^−1^ is the stretching vibration absorption peak of -OH, the absorption peak at 2935 cm^−1^ is the stretching vibration absorption peak of C-H, the absorption peak at 1743 cm^−1^ is the carbonyl absorption peak, the absorption peaks at 1423 cm^−1^ and 1374 cm^−1^ are the deformation absorption peak of = CH2. The absorption peak at 1238 cm^−1^ is the C-O absorption peak of fatty ether, the absorption peaks at 1079 cm^−1^ and 1024 cm^−1^ are the variable Angle vibration absorption peak of alcohol hydroxy-OH, and the absorption peak at 892 cm^−1^ shows the presence of β-type glycosidic bond which indicated that SPs is a β-type polysaccharides [24,25].

The similarity of 10 batches of SP was calculated by the correlation coefficient (R) method and cosine (cos θ) method, and the results are shown in Appendix A. The mean correlation coefficient value and the mean cosine value were 0.981 and 0.994, which indicated that the similarity was high enough to consider that the generated standard FT-IR fingerprint can reflect most of the functional group characteristic features for the SPs.

#### 2.3.2. Chemometric Analysis of FT-IR Fingerprint

Taking the area of the common peaks in the FT-IR fingerprint of SPs as the variable, the Ward method was used for clustering, and the square Euclidean distance method was used as the classification basis. When the discriminant distance was 5, there were three kinds of polysaccharide clusters. Among them, S7, S8, and S10 are the first class, S2 and S3 are the second class, and S1, S4, S5, S6, and S9 are the third class, as shown in Figure 6 A. The samples from Heilongjiang, Jilin, and Inner Mongolia cannot be grouped according to origin. the results showed that there was little correlation between the infrared spectral characteristic of SPs and their area.

In order to compare the structural differences of SPs from different areas, 10 characteristic peaks of the infrared spectrum fingerprints of 10 batches of SP were imported into SPSS 20.0 software for principal component analysis. The results showed that the contribution rate of the first four principal components was 99.918% (>70%), so the first four principal components were selected for evaluation. All the 10 variables were greatly correlated with PC 1 and the peak of wave number 892 cm^−1^ contributed greatly to PC 2, as shown in Appendix A. The PCA score showed that the 10 batches of SPs were divided into three groups, S7, S8, and S10 are the one group, S2 and S3 are the one group, S1, S4, S5, S6, and S9 are another group. The results were consistent with the cluster analysis, as shown in Figure 6B,C.

In order to find the difference markers between different batches of SP, the PLS-DA model was used. The results showed that R^2^X (cum) = 0.999, R^2^Y (cum) = 0.866, and Q^2^ (cum) = 0.506, indicating that the model was stable and reliable. Figure 6D,E shows that SPs can be divided into three groups, S7, S8, and S10 are the one group, S2 and S3 are the one group, S1, S4, S5, S6, and S9 are another group, which is consistent with cluster analysis and PCA analysis. VIP value was used as the screening criteria to obtain the landmark absorption peaks that could distinguish SPs samples from different origins. As can be seen from Figure 6F, VIP values of 892 cm^−1^, 1024 cm^−1^, 1743 cm^−1^, 3389 cm^−1,^ and 2935 cm^−1^ were all greater than 1, indicating that the above characteristic peaks had a great influence on the FT-IR spectrum quality of SPs.

### 2.4. Anti-Allergic Activity of SPs

MTT assay was used to determine the effect of SPs on cell viability of RBL-2H3 cells, and the results showed that the viability of cells cultured in the concentration range of 0.3~1000 μg mL^−1^ SPs were as follows: 78.8%, 93.6%, 90.2%, 100.1%, 92.8%, 97.6%, 89.6%, 93.8%. Compared with the normal control group (NC), there was no statistically significant difference between groups, as shown in Figure 7A. According to the above experimental results, the concentration range of 0.3~500 μg mL^−1^ was selected to investigate the effect of SPs on the activation of β-HEX in RBL-2H3 cells.

The effects of SPs at concentrations of 0.3, 1, 3, 10, 30, 100, and 500 μg mL^−1^ on the release of IgE-mediated degranulation of β-HEX in RBL-2H3 cells were investigated. The results showed that SPs could significantly inhibit the release of β-HEX from RBL-2H3 cells in the concentration range of 0.3~500 μg mL^−1^, and the inhibition rates were 21.8%, 18.3%, 49.9%, 51.6%, 40.5%, 48.1%, 60.0%, respectively, as shown in Figure 7B. The results showed that the high concentration of SPs had a strong general inhibitory ability, but it did not have concentration dependence. Among them the concentration of 500 μg mL^−1^ was the strongest, so this concentration was selected to conduct the study on the inhibition of activation of RBL-2H3 cells by 10 batches of SPs.

The inhibition of β-HEX release from RBL-2H3 cells by SPs derived from 10 batches was carried out at a concentration of 500 μg mL^−1^. The inhibitory rates of β-HEX of the SPs from batches S1 to S10 were 70.9%, 66.4%, 87.1%, 59.3%, 74.5%, 56.1%, 59.0%, 57.2%, 65.3% and 65.6%, respectively, as shown in Figure 7C. The results showed that the SPs extracted from the third and fifth batch (S3 and S5) had the strongest inhibitory effect on the release of activation degranulation β-HEX in RBL-2H3 cells. Both of them were wild *Saposhnikoviae Radix* harvested from Inner Mongolia in the winter. SPs of wild *Saposhnikoviae Radix* harvested from Inner Mongolia but collected in late spring and early summer (S6) had similar efficacy to other SPs. The results of this study suggest that the quality of the *Saposhnikoviae Radix* polysaccharide harvested in winter is better. Geng et al. found that *Saposhnikoviae Radix* polysaccharide could treat allergic rhinitis by down-regulating the levels of IL-4, IL-5, and IgE in the serum of rats and up-regulating the levels of IFN-γ and IL-12 [26]. For the DTH mouse model induced by DNFB, Gao et al. found that *Saposhnikoviae Radix* polysaccharide could inhibit ear swelling, improve thymus index and reduce IgE level in serum of mice, proving that *Saposhnikoviae Radix* polysaccharide has anti-allergic activity [22]. Previous studies have shown that the content of chromogen in *Saposhnikoviae Radix* in early May in spring or mid-October in autumn is higher [27], which indicated that the harvesting season had a definite influence on the quality of *Saposhnikoviae Radix*.

### 2.5. Spectrum-Effect Relationship Analysis

Two data matrices were obtained by dimensionless processing of the peak area of PMP-HPLC fingerprint, the peak area of HPSEC fingerprint, and the inhibition rate of β-HEX of 10 batches of SP according to the averaging formula. The dimensionless data were substituted into the calculation formula of correlation number and correlation degree to obtain the correlation degree between each chromatographic peak and anti-allergic activity and then sorted, as shown in Table 1 and Table 2. Among the six characteristic peaks in the HPLC chromatogram of SP’s monosaccharide composition, the correlation degrees of No. 2, 3, 4, 5, and 6 peaks were all greater than 0.6, indicating that the anti-allergic activity of SPs was jointly played by glucose, rhamnose, galactose, galacturonic acid, and arabinose. The correlation between peak No.2 (rhamnose) and peak No.4 (glucose) was more prominent, indicating that the content of rhamnose and glucose had a greater effect on the anti-allergic activity of SPs. The correlation degree of peak No.1 (mannose) was less than 0.6, the results showed that mannose content had little effect on the anti-allergic activity of SPs. Previous PMP-HPLC analysis showed that rhamnose and galactose were the differential markers of SPs, suggesting that rhamnose and galactose should be paid more attention to in the quality control of SPs.

Among the three characteristic peaks of molecular weight distribution of SPs in the HPSEC chromatogram, the correlation degree of peak 1, peak 2, and peak 3 were all greater than 0.6, indicating that the three parts of different molecular weight in SP played a role together in the anti-allergic activity of SPs, and peak 2 (Mn = 2.50 × 10^6^~3.15 × 10^6^Da) had the highest correlation with the anti-allergic activity of SPs.

Multiple fingerprints of *Saposhnikoviae Radix* polysaccharides were prepared by PMP-HPLC, HPSEC, and FT-IR fingerprints. PMP-HPLC fingerprint showed that SPs were composed of mannose, rhamnose, galacturonic acid, glucose, galactose, and arabinose. HPSEC fingerprint indicates that SPs are composed of three polysaccharide fragments, and the Mn of the three polysaccharide fragments were ranging from 8.67 × 10^6^~9.56 × 10^6^ Da, 2.50 × 10^6^~3.15 × 10^6^ Da, and 1693~3999 Da respectively. Two monosaccharides (rhamnose and galactose), the polysaccharide fragment Mn = 8.67 × 10^6^~9.56 × 10^6^ Da, and the FT-IR absorption peak of 892 cm^−1^ can be used as the quality control markers of SPs. The results showed that SPs had anti-allergic activity. The spectrum-effect relationship model showed that the monosaccharide composition and molecular weight were related to the anti-allergic activity of the SPs. In the future, we will consider further anti-allergic activity experiments and spectrum-effect relationship studies, so as to obtain a more reliable active quality marker of *Saposhnikoviae Radix* polysaccharide.

## 3. Materials and Methods

### 3.1. Materials and Cell

Ten batches of *Saposhnikoviae Radix* slices samples were collected from various varieties and origins in different regions, batches, and seasons, such as Heilongjiang (S1, S2), Neimenggu (S6, S10), Hebei (S4), Jilin (S7, S8, S9), showed in Appendix A. Rat RBL-2H3 cells were purchased from Shanghai Ze Ye Biotechnology Co., Ltd. (Shanghai, China).

### 3.2. Preparation of Polysaccharide Extracts

Referring to the extraction method of Gao et al. [22], 600 g of *Saposhnikoviae Radix* slices sample were refluxed by 10 times deionized water for 1.5 h, extracted 3 times, and merged with the liquid medicine. The extract was concentrated to a crude drug content of 1 g mL^−1^, add 95% ethanol until the solution contains 80% alcohol at room temperature, and alcohol precipitation was at 4 °C for 12 h. Next, discard the supernatant, the deposit was washed with 80% ethanol 3 times and dried in a vacuum drying oven, then obtained the raw polysaccharide. The raw polysaccharide was purified by Sevag method, dialysis method (molecular weight cut-off 3500 Da) to remove protein and small molecular impurities, and then freeze-dried to obtain refined polysaccharides from *Saposhnikoviae Radix* for subsequent experiments.

### 3.3. Characterization of SP and Multiple Fingerprint Profiles

The PMP pre-column derivation HPLC (PMP-HPLC) was used to determine the monosaccharide composition of SPs [28,29]. Ten batches of polysaccharide samples (20 mg) were hydrolyzed in sealed vials with 4 mL of 2 mol L^−1^ trifluoroacetic acid (TFA) respectively, hydrolyzed in the blast drying oven at 110 °C for 6 h. Centrifuged the hydrolysate at 5000 rpm for 10 min after cooling, take the supernatant and add 1 mL of methanol, then rotary evaporated and repeated until the TFA is exhausted. Then dissolved the dried hydrolysates with 1 mL of deionized water. In the next moment, the 500 µL of hydrolysate was mixed with 200 µL of 0.3 mol L^−1^ NaOH and 200 μL of methanolic solution of PMP (0.5 mol L^−1^). The reaction was set at 70 °C for 1 h. The reaction solution was neutralized with 200 μL of HCl (0.3 mol L^−1^) when the solution cooled down to room temperature, and eventually extracted with 1 mL of chloroform five times. The final aqueous phase was collected and filtered through a 0.22 μm filter membrane prepared for HPLC analysis. The reference standard contains 8 different kinds of monosaccharides (d-anhydrous glucose, d-arabinose, l-Rhamnose, d-galactose, d-galacturonic acid, d-mannose, d-ribose, l-fucose) was processed by using the same method. A total of 10 µL of the derivatives (standards and polysaccharides samples) were analyzed by LC-20AT Shimadzu Liquid Chromatographs equipped with a ZORBAX Eclipse XDB-C18 HPLC column (4.6 mm × 250 mm, 5 μm, Agilent, Santa Clara, CA, USA). The column temperature was operated at room temperature and the ultraviolet detector wavelength was set to 245 nm. The mobile phase consisted of 0.1 mol L^−1^ phosphate-buffered saline (pH 6.7) and acetonitrile (*v*/*v* = 82:18) with a flow rate of 0.8 mL min^−1^.

High-performance size exclusion chromatography (HPSEC) analysis was used to determine the molecular weight distribution of SPs [30,31]. The analysis was performed on Waters 1525 High-performance liquid chromatography system coupled 2412 Waters refractive index detector. Using a TSK-GMPWxl column (7.8 mm × 300 mm, 13 µm, Tosoh Corporation, Tokyo, Japan), 10 μL of the sample was injected and eluted at 40 °C with deionized water at a flow rate of 0.5 mL min^−1^. Calculate the molecular weight of polysaccharides from the dextran molecular weight standard (China National Institute for Food and Drug Control, Beijing, China).

FT-IR fingerprint analysis was used to determine the characteristic structure of SPs [32]. The IR spectra of polysaccharides samples were recorded with Fourier transform infrared spectrometer (Nicolet iS10, Thermo, Waltham, MA, USA). Ten batches of polysaccharides samples of about 1 mg were weighed. Each sample was ground with spectroscopic grade 100 mg KBr powder and then pressed into pellets for FT-IR measurement in the frequency range of 4000–400 cm^−1^.

### 3.4. Chemometrics Analysis

The similarity evaluation was performed by the TCM chromatographic fingerprints (Version 2012) (Chinese Pharmacopoeia Commission, Beijing, China) recommended by the SFDA of China. Similarities analysis was evaluated by methods including correlation coefficient (R) and angle cosine (cos θ). The formulas for calculating similarity were as follows:$$R=\frac{\sum_{i=1}^{n}\left(X_{i}-{\bar{X}}_{i}\right)\left(Y_{i}-{\bar{Y}}_{i}\right)}{\sqrt{\sum_{i=1}^{n}{\left(X_{i}-{\bar{X}}_{i}\right)}^{2}{\left(Y_{i}-{\bar{Y}}_{i}\right)}^{2}}}\;\cos\;\theta =\frac{\sum_{i=1}^{n}X_{i}Y_{i}}{\sqrt{\sum_{i=1}^{n}X_{i}^{2}}\sqrt{\sum_{i=0}^{n}Y_{i}^{2}}}$$

Hierarchical cluster analysis (HCA) and Principal component analysis (PCA) were performed by using the statistics software of Statistical Package for Social Sciences (SPSS), version 20.0 (IBM Company, New York, NY, USA). In this study, the Ward method was used for clustering, and the Square Euclidean distance method was used as the classification basis for the HCA of samples. SMICA 14.1 software was used for further partial least squares-discriminant analysis (PLS-DA).

### 3.5. Determination of SPs Anti-Allergic Activity

The cytotoxicity of SPs on Rat RBL-2H3 cells was determined by MTT assay [33,34]. Take the logarithmic growth stage cells to make cell suspension and adjust cell concentration to 5 × 10^4^/mL. The blank group, control group, and 8 medication groups were set on the 96-well plate, with 6 duplicate holes in each group. 100 μL of cell suspension was added to each well in the control and administration groups, and a complete medium was added to the blank group. After cell inoculation, 96-well plates were transferred to a 37 °C and a 5% CO_2_ incubator for 24 h. The original medium was discarded, and the drug groups at different concentrations (0.3, 1, 3, 10, 30, 100, 500, and 1000 μg mL^−1^) were added with 100 μL SPs solution per well. The blank group and control group were added with the same amount of basic medium and incubated for 24 h. Discard the drug solution, add 10% MTT medium to each well, incubate for 4 h, discard the supernatant, add 100 μL DMSO, shake the shaker for 10 min, and detect OD value at 570 nm by enzyme labeling instrument (BioTek Instruments, Inc., Winooski, VT, USA).

Evaluation of the anti-allergy activity of SPs was finished by degranulation of RBL-2H3 cells mediated IgE [35]. The normal group, model group, medication group, and negative control group were set on the 24-well plate, with three replicate wells in each group. Add 1000 μL cell suspension (cell concentration, 5 × 10^4^/mL) to each well, transfer to 37 °C and 5% CO_2_ incubator for 24 h. After cells adhered to the wall, the DNP-IgE working solution (final concentration: 0.25 μg mL^−1^) was added to the model group and each administration group, and an equal amount of DMEM basic medium was added to the normal group and negative control group, incubated for 12 h. After the cells were sensitized for 12 h, discarded the medium, added different concentrations (0.3, 1, 3, 10, 30, 100, 500 μg mL^−1^) of 200μL of SPs solution to each well of the drug group and the negative control group, then incubated for 2.5 h. The normal group and the model group were added with the same amount of DMEM basic medium. Discard the supernatant and rinsed the cells 3 times with PBS buffer. DNP-BSA (0.25 μg mL^−1^, 200 μL) was added to each well of the model group and medication group. The same amount of PIPES buffer was added into each well of the normal group and negative control group. After incubated for 1 h, the release of β-HEX was stimulated.

The reaction was terminated in an ice bath for 10 min, and the supernatant was collected by centrifugation. Then, take 50 μL supernatant per well to 96-well plate, add 50 μL chromogenic solution and react in constant temperature incubator for 1 h, finally added 200 μL stop solution to stop the reaction. The OD value was detected by a microplate reader at 405 nm. The same method was used to determine the effect of different batches of SPs on the release of β-HEX. The calculation formula for β-HEX inhibition rate is as follows:$$\beta \text{-}\mathrm{HEX}\;\mathrm{Inhibition}\;\mathrm{rate}\;\%=\;(1-\frac{\mathrm{OD}\;\mathrm{value}\;\mathrm{of}\;\mathrm{SP}\;\mathrm{group}\;-\;\mathrm{OD}\;\mathrm{value}\;\mathrm{of}\;\mathrm{SP}\text{-}\mathrm{nc}\;{\mathrm{group}}_{D}-\;\mathrm{OD}\;\mathrm{value}\;\mathrm{of}\;\mathrm{Control}\;\mathrm{group}}{\mathrm{OD}\;\mathrm{value}\;\mathrm{of}\;\mathrm{Model}\;\mathrm{group}\;-\;\mathrm{OD}\;\mathrm{value}\;\mathrm{of}\;\mathrm{Control}\;\mathrm{group}})\times 100\%$$

### 3.6. Spectrum-Effect Relationship Analysis

Based on the fingerprints and pharmacodynamics data obtained previously, the grey correlation analysis was used to establish the spectrum-effect relationship and screen the key variables. The β-HEX inhibition rate was selected as the reference series, and the peak areas of the common peaks were selected as the compared series. After the normalization of the original data by “Z-SCORE”, the gray relational coefficients for each common peak were obtained [36].

### 3.7. Statistical Analysis

All Data were shown as mean ± standard deviation (SD). The GraphPad Prism 6.0 software was used for analysis and processing. T-test and One-way ANOVA were used to evaluate the significant difference between the experimental groups and the control group. *p* < 0.05 indicated that the results were significantly different, and *p* < 0.01 indicated that the difference was extremely significant.

## 4. Conclusions

Fingerprint analysis is a comprehensive and effective method to evaluate the authenticity and quality of traditional Chinese medicine and natural products. In this study, PMP-HPLC, HPSEC and FT-IR fingerprints were used to establish the multiple fingerprints of 10 batches of SP. The similarity of PMP-HPLC, HPSEC, and FT-IR fingerprints were above 0.953, 0.998, and 0.959, respectively, indicating that the three control chromatography could all reflect the basic characteristics of SPs. Cluster analysis, principal component analysis, and partial least squares discriminant analysis were used to analyze the main markers of the quality of SPs. Among them, the contents of rhamnose and galactose had a great influence on the quality of the SPs. The polysaccharide fragment Mn = 8.67 × 10^6^~9.56 × 10^6^ Da, Mn = 2.50 × 10^6^~3.15 × 10^6^ Da, and infrared structure, the 892 cm^−1^ peak of the β -configuration can be used as an index to evaluate the quality of SPs.

Evaluation of the anti-allergic activity of SPs was finished by degranulation of RBL-2H3 cells mediated IgE. The results showed that the anti-allergic activity of SPs was the best at the concentration of 500 μg mL^−1^, and all 10 batches of SP had a good anti-allergic effect at the concentration of 500 μg mL^−1^, among which the third batch (S3) and the fifth batch (S5) of SP from the Inner Mongolia wild *Saposhnikoviae Radix* had the strongest activity. The spectral effect relationship was established between the fingerprints and the anti-allergic activity of SPs by grey correlation analysis. The results showed that glucose and rhamnose played a major role in the anti-allergic effect of the SPs and the polysaccharide fragments with molecular weight Mn = 8.67 × 10^6^~9.56 × 10^6^ Da and Mn = 2.50 × 10^6^~3.15 × 10^6^ Da had the greatest correlation with the anti-allergic pharmacodynamics.

A variety of fingerprints established in this study can complete the quality control of SPs from multiple perspectives, overall quality, and comprehensive characteristics, and the spectral-effect relationship can evaluate the quality of SPs from its efficacy, so that the quality control method of SPs could have more practical significance. In contrast to studies of the spectral-effect relationship of other natural extracts, polysaccharide spectral-effect relationship studies relate the efficacy to certain monosaccharides rather than chemical compositions. However, monosaccharide usually has no pharmacological activity, and there are few studies about the effect of monosaccharide composition on the efficacy of polysaccharides. This leads to the lack of evidence to support the corresponding conclusions in the spectrum-effect analysis of polysaccharides. Therefore, the application of spectral effect relationship analysis was greatly limited in the field of polysaccharides, and more studies are needed that research the effect of monosaccharide composition on spatial structure and pharmacological activity.

## Figures and Tables

**Figure 1 molecules-27-05278-f001:**
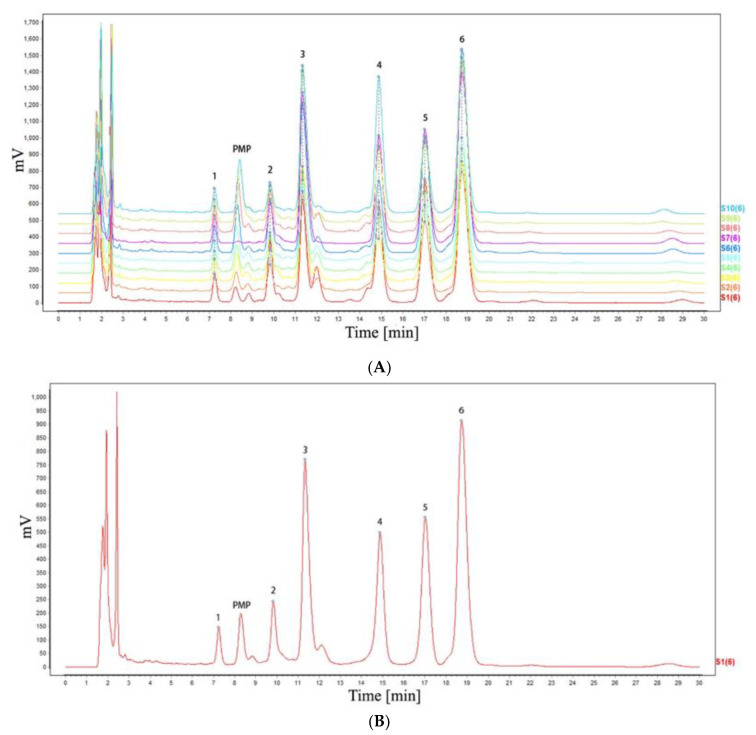
The PMP-HPLC fingerprints. (**A**) the PMP-HPLC fingerprints of different SPs (**B**) the PMP-HPLC standard referential fingerprint of SPs.

**Figure 2 molecules-27-05278-f002:**
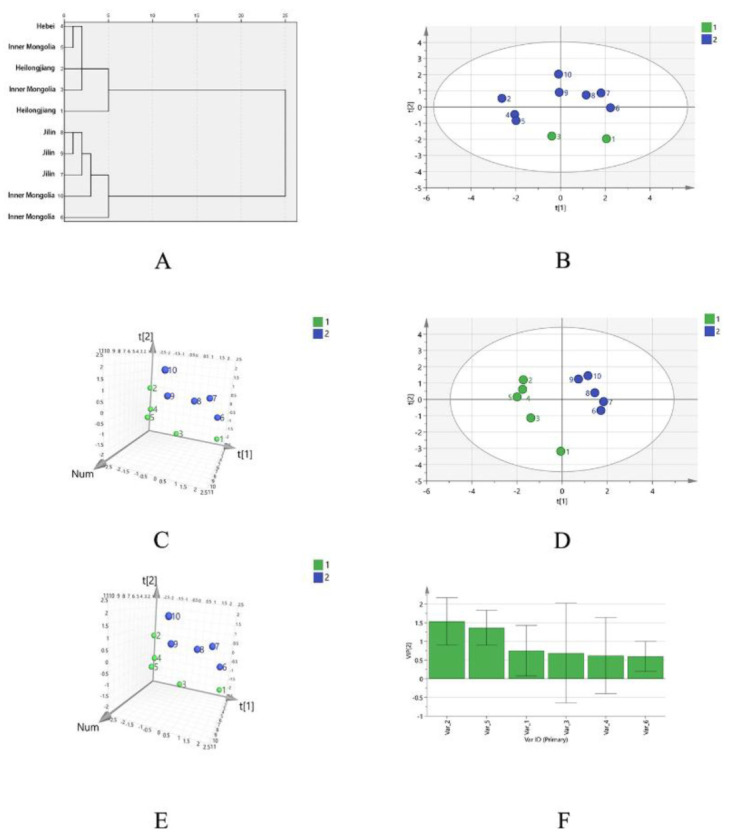
The dengrogram (**A**), the PCA score plot (**B**), and the loading plot (**C**) of PMP-HPLC fingerprints of different SPs. The PLS-DA score plot (**D**), loading plot (**E**), and the VIP predicted value plot (**F**) of PMP-HPLC fingerprints of different SPs.

**Figure 3 molecules-27-05278-f003:**
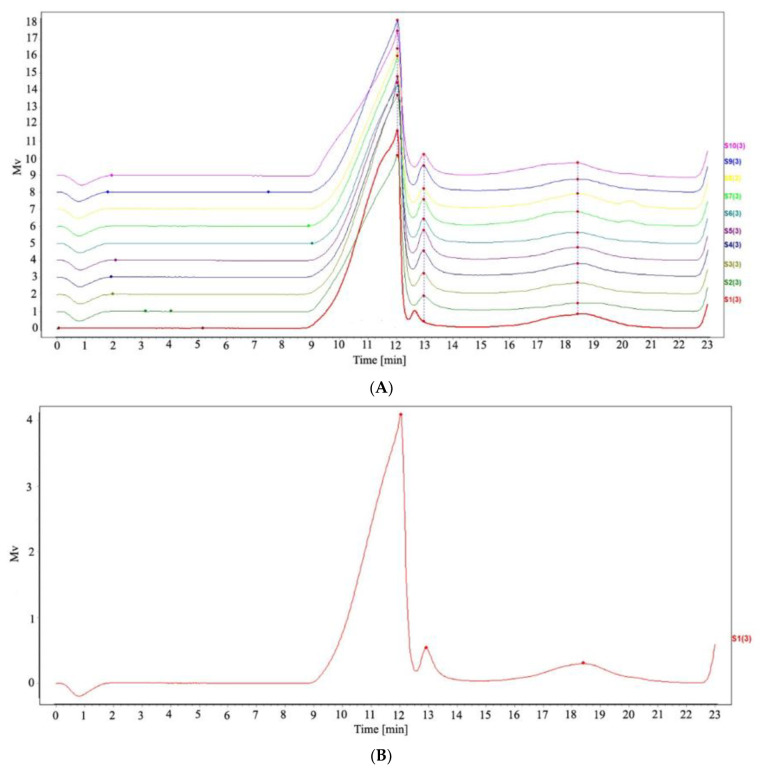
The HPSEC fingerprints. (**A**) the HPSEC fingerprints of different SPs (**B**) the HPSEC standard referential fingerprint of SPs.

**Figure 4 molecules-27-05278-f004:**
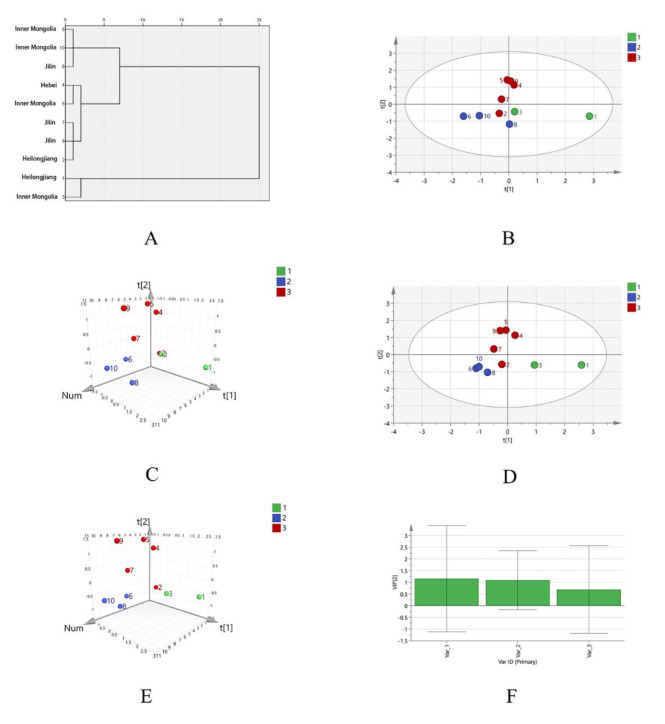
The dengrogram (**A**), the PCA score plot (**B**) and loading plot (**C**) of HPSEC fingerprints of different SPs. The PLS-DA score plot (**D**), loading plot (**E**) and the VIP predicted value plot (**F**) of HPSEC fingerprints of different SPs.

**Figure 5 molecules-27-05278-f005:**
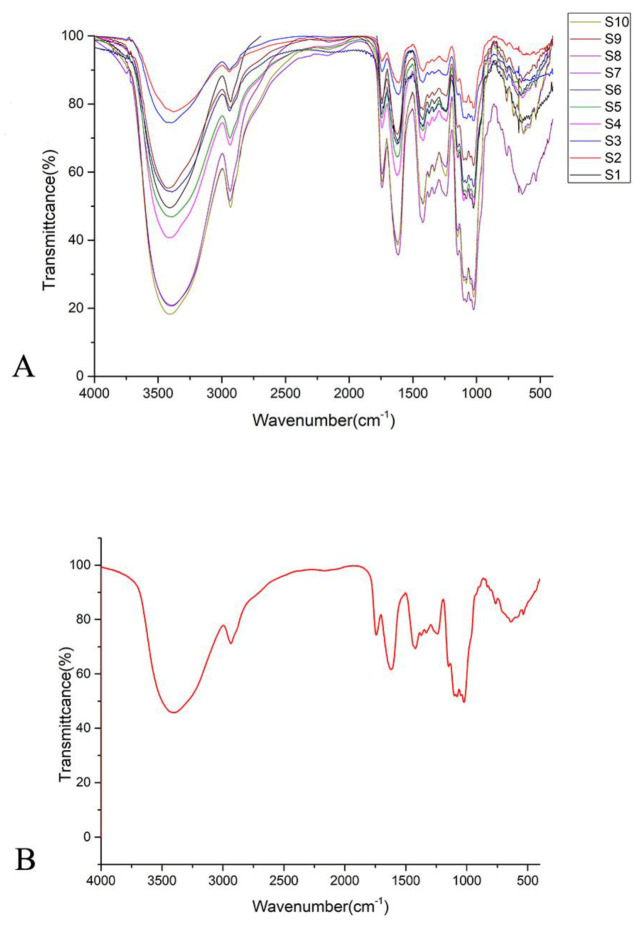
The FT-IR fingerprints. (**A**) the FT-IR fingerprints of different SPs (**B**) the FT-IR standard referential fingerprint of SPs.

**Figure 6 molecules-27-05278-f006:**
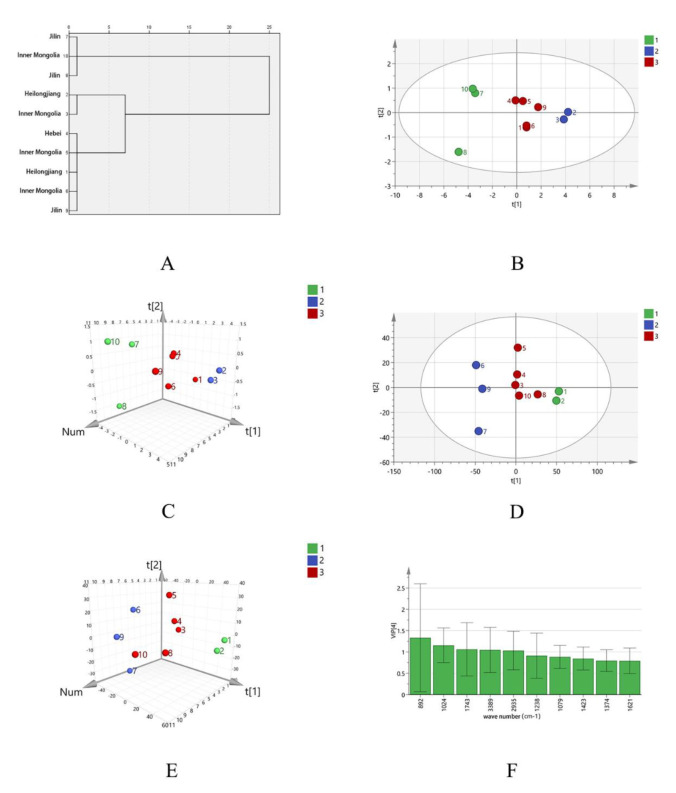
The dengrogram (**A**), the PCA score plot (**B**) and loading plot (**C**) of FT-IR fingerprints of different SPs. The PLS-DA score plot (**D**), loading plot (**E**), and the VIP predicted value plot (**F**) of FT-IR fingerprints of different SPs.

**Figure 7 molecules-27-05278-f007:**
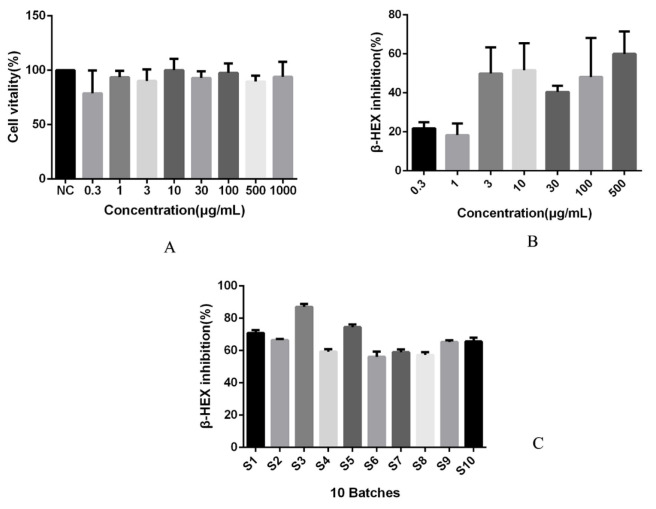
The effect of SPs on cell viability (**A**). The Inhibitory effect of different concentrations (**B**) and batches (**C**) of SPs on β-HEX release, the inhibition rate of the control group was zero.

**Table 1 molecules-27-05278-t001:** Correlation between PMP-HPLC fingerprints and anti-allergic activity of SPs.

Peak	Correlation (r)	Monosaccharide
4	0.763	Glucose
2	0.716	Rhamnose
5	0.700	Galactose
3	0.672	Galacturonic acid
6	0.657	Arabinose
1	0.586	Mannose

**Table 2 molecules-27-05278-t002:** Correlation between HPSEC fingerprints and anti-allergic activity of SPs.

Peak	Correlation (r)	Number Average Molecular Weight (Mn)
2	0.767	2,245,648 Da
1	0.694	10,338,080 Da
3	0.679	2295 Da

## Data Availability

The data presented in this study are available in article and Appendix A.

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
