# Peer review of "Multiple Fingerprints and Spectrum-Effect Relationship of Polysaccharides from Saposhnikoviae Radix"

_molecules, 2022, doi:10.3390/molecules27165278_

Round 1

Reviewer 1 Report

The authors prepare a polysaccharide extract (PS) from ten different samples of a plant species that has been used as a traditional medicine and compare various aspects of this extract using principle component analysis.

There is a nice approach to the work, but I do have some concerns, detailed below:

1)  It is never explained what part of the plant is used.  "Slices"?  The traditional preparation and use of the plant must be introduced, and then why the PS so extracted in this work is relevant to the medicinal properties quoted. See also 2) below.

2) The naming is not clear...I had to look up the literature to discover: "Saposhnikovia divaricata (Turcz.) Schischk., a perennial herb belonging to the family Umbelliferae, is widely distributed in Northeast Asia. Its dried root (Radix Saposhnikoviae) is used as a Chinese herbal medicine for the treatment of immune system, nervous system, and respiratory diseases" 

3) Why are the plant names not in italic?

4) I don't believe this is "multiple fingerprints"  You are collecting a single fingerprint for ten different materials.

5) My major concern is the reproducibility and baseline data.  I can find no reference to the root material being extracted, even in duplicate.  So how much variation and what error is present in each sample?  Is the variation we are seeing in such a small sample set actually due to possible variation in the extraction preparations, or between sliced material?  Were multiple slices from each location combined to average a locale? 

This is a bit of a worry as really each data point should be:

a) several slices (to cover between-plant variation) 

b) completed in triplicate for each location using a) and the error reported

6) I don't know what spectrum-effect is. This is an odd term.

7) The English language use is ok, but could do with some minor improvement in the text, but needs major work in the experimental.  There are the odd typo and unclear statements throughout.

8) Lines 102-105 are clumsy and need reworking.

9) The SEC-HPLC is, and I am not sure how to really put this, except; wrong.  The peak shape is not consistent with a molecular weight distribution running from an SEC.  SEC Mw distributions do not culminate in a sharp peak and then drop off so suddenly ...if anything they tail as there is a greater heterogeneity in the smaller molecules (rather a wider range of hydrodynamic volume giving the appearance of a more heterogeneous distribution).   Consider the SEC reported in https://doi.org/10.1016/j.carbpol.2021.119081 (which are not PS but heparan sulfates but it shows the kind of peak shapes one would normally encounter). In addition, for SEC data, Mn & Mw data should be calculated.  And, in the experimental, exactly how this was completed needs to be explained - I note that with the peak shape observed dispersity would end up as a very odd/nonsensical number.

10) There is no information on the yield of PS from the extraction process.

11) Attention is required with respect to significant figures.  If you can generate 3 sig.fig. for SEC you are doing very well, and similarly for % values.

12) This is a personal comment - I am not convinced that IR data can be used for anything constructive in this type of analysis. It is particularly difficult when the amount of water that may be present is considered and if a spectroscopic analysis is required then the current state of the art is NMR.

Author Response

Dear Reviewer,

Thank you for your comments on our article entitled " Multiple fingerprints and spectrum-effect relationship of polysaccharides from Saposhnikoviae Radix" (Manuscript ID: molecules-1826140). These comments are very valuable and helpful for us to revise the manuscript. Accordingly, we carefully studied the comments and revised the manuscript. Some of the revised fonts were marked in red. Itemized responses to the comments are listed below the letter. We hope the modification can meet the requirements of reviewers. Looking forward to your reply.

Thank you and best wishes.

Yours sincerely,

Guang Xu.

Your original comments and my response are as follows

The authors prepare a polysaccharide extract (PS) from ten different samples of a plant species that has been used as a traditional medicine and compare various aspects of this extract using principle component analysis.

There is a nice approach to the work, but I do have some concerns, detailed below:

1)  It is never explained what part of the plant is used.  "Slices"?  The traditional preparation and use of the plant must be introduced, and then why the PS so extracted in this work is relevant to the medicinal properties quoted. See also 2) below.

Response: Thank you for your suggestion. According to the Chinese Pharmacopoeia, the medicinal part of Saposhnikoviae Radix is dry root, and the dry root was cut into sheets for clinical use. According to the 2) below, the traditional clinical use of Saposhnikoviae Radix was introduced, modified in Lines 32-33. The traditional use of Saposhnikoviae Radix is to prepare it as a decoction. The extraction method of Saposhnikoviae Radix polysaccharide was based on traditional clinical methods and previous research literature, I have further supplemented the corresponding references in the method section, modified in Lines 370.

2) The naming is not clear...I had to look up the literature to discover: "Saposhnikovia divaricata (Turcz.) Schischk., a perennial herb belonging to the family Umbelliferae, is widely distributed in Northeast Asia. Its dried root (Radix Saposhnikoviae) is used as a Chinese herbal medicine for the treatment of immune system, nervous system, and respiratory diseases"

Response: Thanks for your review. I'm sorry for the confusion and I'll explain it as follow. There are three names in the article, all from the Chinese Pharmacopoeia. “Fangfeng” is the pinyin spelling of the official Chinese name stipulated by the pharmacopoeia, “Saposhnikoviae Radix” is the official English name stipulated by the pharmacopoeia, and “Saposhnikovia divaricataTurcz.Schischk” is the Latin name of the base plant according to the pharmacopoeia. I have revised the original text to avoid possible confusion, modified in Lines 31-33, I also use “Saposhnikoviae Radix” for the following descriptions.

3) Why are the plant names not in italic?

Response: Thanks for your comment. It has been revised in the full text.

4) I don't believe this is "multiple fingerprints"  You are collecting a single fingerprint for ten different materials.

Response: Thank you for your review. The multiple fingerprints of polysaccharides, is using various analytical methods, including HPLC, IR, UV, GPC, etc. to obtain single fingerprint reflecting different physical and chemical characteristics of polysaccharides, and integrate the results of single fingerprints to form multiple fingerprints of polysaccharides. At the same time, using stoichiometric methods, such as SA, PCA and HCA, the similarity and difference of polysaccharide structures can be judged. Similar multiple fingerprints studies of polysaccharides from Sarcandra glabra, Flammulina velutipes and Zishen Yutai Pills have been carried out before[1-3].

5) My major concern is the reproducibility and baseline data.  I can find no reference to the root material being extracted, even in duplicate.  So how much variation and what error is present in each sample?  Is the variation we are seeing in such a small sample set actually due to possible variation in the extraction preparations, or between sliced material?  Were multiple slices from each location combined to average a locale? 

This is a bit of a worry as really each data point should be:

  1. a) several slices (to cover between-plant variation) 
  2. b) completed in triplicate for each location using a) and the error reported

Response: Thank you for your thoughtful suggestion. The repeatability of the experiment was guaranteed, and the variation in the experiment were due to the harvesting season, the harvesting area, and the wild and planted varieties. The samples used in the study were commercially available TCM decoction pieces collected from representative producing areas, which met the quality control standards stipulated in the Chinese Pharmacopoeia. Before the experiment, Professor Yaojun Yang, an expert in the identification of Traditional Chinese medicine, identified the samples to ensure that the samples used in the experiment were representative of the region. In addition, the amount of raw materials used for extraction is relatively large, so there will be no difference caused by different sample locations. The extraction methods of samples were completely consistent, and I further improved the information related to extraction in the method part, modified in Lines 370-371.

6) I don't know what spectrum-effect is. This is an odd term.

Response: Thanks for your review. The establishment of spectrum-effect relationship is based on data analysis method to correlate chromatographic results with pharmacodynamic indexes, and then screen out the components that have a great influence on pharmacodynamic indexes. The commonly used data analysis methods include grey correlation degree method, partial least squares regression analysis, correlation analysis, hierarchical cluster analysis, etc. This kind of correlation analysis has been continuously used in recent years[4-7].

7) The English language use is ok, but could do with some minor improvement in the text, but needs major work in the experimental.  There are the odd typo and unclear statements throughout.

Response: Thank you for your review. Textual improvements have been made to the full text, and typos and unclear statements have been eliminated.

8) Lines 102-105 are clumsy and need reworking.

Response: Thanks for your comment. Line 102-105 have been rewritten in accordance with your comments.

9) The SEC-HPLC is, and I am not sure how to really put this, except; wrong.  The peak shape is not consistent with a molecular weight distribution running from an SEC.  SEC Mw distributions do not culminate in a sharp peak and then drop off so suddenly ...if anything they tail as there is a greater heterogeneity in the smaller molecules (rather a wider range of hydrodynamic volume giving the appearance of a more heterogeneous distribution).   Consider the SEC reported in https://doi.org/10.1016/j.carbpol.2021.119081 (which are not PS but heparan sulfates but it shows the kind of peak shapes one would normally encounter). In addition, for SEC data, Mn & Mw data should be calculated.  And, in the experimental, exactly how this was completed needs to be explained - I note that with the peak shape observed dispersity would end up as a very odd/nonsensical number.

Response: Thank you for your very critical comments. The shape of the peak may be caused by the large molecular weight. The molecular weight of the polymer in the reference you mentioned is about 5 digits, while the main peak of the Saposhnikoviae Radix polysaccharide we studied, that is, the less typical peak type, is about 7 digits. We are very sorry that we did not submit the relevant molecular weight data due to our negligence. We submitted Table S 3.1 in the supplementary materials, which contains the Mn, Mw, Mz and DPI data of 10 batches of SP.

10) There is no information on the yield of PS from the extraction process.

Response: Thanks for your review. This study mainly focused on the multiple fingerprints, stoichiometric analysis and spectrum-effect analysis of polysaccharides. So we didn't put the yield of polysaccharide in the paper, and the average yield of our 10 batches of polysaccharide was about 3.42%.感谢您的审核。本研究主要对多糖的多重指纹图谱、化学计量分析和谱效分析进行了研究。

11) Attention is required with respect to significant figures.  If you can generate 3 sig.fig. for SEC you are doing very well, and similarly for % values.

Response: Thanks for your comment. The figures have been modified according to your suggestion and the quality of the picture has been improved.

12) This is a personal comment - I am not convinced that IR data can be used for anything constructive in this type of analysis. It is particularly difficult when the amount of water that may be present is considered and if a spectroscopic analysis is required then the current state of the art is NMR.

Response: Thanks for your review. For quality control studies of polysaccharides, monosaccharide composition and molecular weight are usually the main ones, and infrared structure data are used to supplement them. NMR can provide the glycosidic bond configuration and the type of monosaccharide residues of polysaccharides, and infrared spectroscopy can provide the information of functional groups of polysaccharides. For the structural characterization of polysaccharides, NMR is more suitable for providing more information. However, this paper mainly focuses on the quality control of polysaccharides. NMR analysis is expensive, complicated and has a higher threshold of data analysis. Therefore, NMR is so it is not universal and not suitable for the quality control of polysaccharides. Compared with NMR, infrared spectroscopy is simple, quick, cheap, and can provide considerable structural information, which is more suitable for the quality control of polysaccharides. In fact, most of the other studies using multiple fingerprints for quality control of polysaccharides have also used infrared spectroscopy for quality control[1, 3, 8].

  1. Li, H.; Cao, J.; Wu, X.; Deng, Y.; Ning, N.; Geng, C.; Lei, T.; Lin, R.; Wu, D.; Wang, S.; Li, P.; Wang, Y., Multiple fingerprint profiling for quality evaluation of polysaccharides and related biological activity analysis of Chinese patent drugs: Zishen Yutai Pills as a case study. Journal of Ethnopharmacology 2020, 260, 113045.
  2. Li, H.; Gong, X.; Wang, Z.; Pan, C.; Zhao, Y.; Gao, X.; Liu, W., Multiple fingerprint profile and chemometrics analysis of polysaccharides from Sarcandra glabra. Int J Biol Macromol 2019, 15, (123), 957-967.
  3. Dong, Y.; Pei, F.; Su, A.; Sanidad., K. Z.; Ma, G.; Zhao, L.; Hu, Q., Multiple fingerprint and fingerprint-activity relationship for quality assessment of polysaccharides from Flammulina velutipes. Food Chem Toxicol 2020, 135, 110944.
  4. Chen, X. Y.; Gou, S. H.; Shi, Z. Q.; Xue, Z. Y.; Feng, S. L., Spectrum-effect relationship between HPLC fingerprints and bioactive components of Radix Hedysari on increasing the peak bone mass of rat. J Pharm Anal 2019, 9, (4), 266-273.
  5. Nijat, D.; Lu, C. F.; Lu, J. J.; Abdulla, R.; Hasan, A.; Aidarhan, N.; Aisa, H. A., Spectrum-effect relationship between UPLC fingerprints and antidiabetic and antioxidant activities of Rosa rugosa. J Chromatogr B Analyt Technol Biomed Life Sci 2021, 1179, 122843.
  6. Qiao, R.; Zhou, L.; Zhong, M.; Zhang, M.; Yang, L.; Yang, Y.; Chen, H.; Yang, W.; Yuan, J., Spectrum-effect relationship between UHPLC-Q-TOF/MS fingerprint and promoting gastrointestinal motility activity of Fructus aurantii based on multivariate statistical analysis. J Ethnopharmacol 2021, 279, 114366.
  7. Ji-Heng Wu; Yue-Ting Cao; Hong-Ye Pan; Wang, L.-H., Identification of Antitumor Constituents in Toad Venom by Spectrum-Effect Relationship Analysis and Investigation on Its Pharmacologic Mechanism. molecules 2020, 25, 4269.
  8. Sun;, X.; Zhao;, Q.; Si;, Y.; Li;, K.; Zhu;, J.; Gao;, X.; Liu, W., Bioactive structural basis of proteoglycans from Sarcandra glabra based on spectrum-effect relationship. Journal of Ethnopharmacology 2020, 259, 112941.

Reviewer 2 Report

Dear authors,

The manuscript entitled " Multiple fingerprints and spectrum-effect relationship of polysaccharides from Saposhnikoviae Radix” used pre-column derivatization high performance liquid chromatography (PMP-HPLC), Fourier transform infrared spectroscopy (FT-IR) and high performance size exclusion chromatography (HPSEC) were used to prepare the multiple fingerprints of 10 batches of polysaccharides from Saposhnikoviae Radix with different production areas and harvest time, and the chemometrics analysis was performed. It presents scientific relevance for the area of Pharmacy, Chemistry, Natural products, Biology and, others area. The language (English) is satisfactory (I suggest the final revision)! However, you need to change some details/information in the title, abstract, Methods, results, discussion and conclusions. I request information on the procedures and interpretation of the results obtained.

Title: Adequate! But, Saposhnikoviae Radix must be written in italics, respecting the genus and species. To review throughout the manuscript, if necessary.

Abstract: Adequate, but I suggest at the end of the abstract, to highlight the "innovative" proposal of the method.

* Introduction section: It is well written, but I suggest:

- Genus and species must be written in italics!

- I suggest inserting a paragraph on the analytical and characterization techniques used.

* Results and discussion section

- Pages 2-3, in “2.1.1. PMP-HPLC Fingerprint and Similarity Analysis” section: I suggest expanding the discussions, focusing on the optimization of the analytical parameters of the LC method, as well as the analytical validation.

- Pages 11-12, in “2.4. Anti-allergic Activity of SPs” section: Are there other authors in the literature who have determined these substances (or others in the group) in species for comparison? I suggest expanding the discussions!

- I suggest, at the end of the "results and discussion", to write a paragraph summarizing the findings and their impacts on the research proposal.

* Material and Methods section:

- The methodological proposals are coherent, but it needs to improve, especially in the analytical aspects.

- Page 13, in “3.1. Materials and Cell” section: How long were the samples stored until the time of analysis?

- Page 13, in “3.2. Preparation of Polysaccharide Extracts” section: Did you follow any references? If yes, enter!

- Page 13, line 357, in “3.3. Characterization of SP and multiple fingerprint profiles” section: To replace “mol/L” by “mol L -1”. To review throughout the manuscript, if necessary.

- Page 14, line 375, in “3.3. Characterization of SP and multiple fingerprint profiles” section: To replace “mL/min” by “mL min -1”. To review throughout the manuscript, if necessary.

- Page 15, in “3.5. Determination of SPs Anti-allergic Activity” section: Why was the Rat RBL-2H3 lineage selected? About the MTT assay, how long were the cells observed? Did you follow any references? If yes, enter! Were SP cytotoxicity assays performed on healthy cells?

- Page 15, in “3.5. Determination of SPs Anti-allergic Activity” section: In line 411, to replace “μg/mL” by “μg mL-1”. To review throughout the manuscript, if necessary. In lines 417-437: long paragraph! I suggest splitting the text!

- Page 15, in “3.6. Spectrum-effect relationship analysis” section: I suggest expanding the information about the correlation analysis, as well as indicating a reference!

What are the analytical validation parameters used for methods? Has the proposed method been validated? If so, which protocol / guidelines did you follow? What are the validation parameters studied? Precision, accuracy, LOD, LOQ, robustness, etc. What concentration levels are used to assess accuracy? I suggest detailing the proposed method in more detail...

* Conclusion section

- Adequate! Long paragraph! I suggest splitting the text! Also, I suggest to indicate disadvantages/limitations of the method and the study!

* Tables and Figures: Adequate!

* Supplementary data: Adequate!

* References: Please, check if the references are in accordance with the journal's rules.

Author Response

Dear Reviewer,

Thank you for your comments on our article entitled " Multiple fingerprints and spectrum-effect relationship of polysaccharides from Saposhnikoviae Radix" (Manuscript ID: molecules-1826140). These comments are very valuable and helpful for us to revise the manuscript. Accordingly, we carefully studied the comments and revised the manuscript. Some of the revised fonts were marked in red. Itemized responses to the comments are listed below the letter. We hope the modification can meet the requirements of reviewers. Looking forward to your reply.

Thank you and best wishes.

Yours sincerely,

Guang Xu.

Your original comments and my response are as follows

The manuscript entitled " Multiple fingerprints and spectrum-effect relationship of polysaccharides from Saposhnikoviae Radix” used pre-column derivatization high performance liquid chromatography (PMP-HPLC), Fourier transform infrared spectroscopy (FT-IR) and high performance size exclusion chromatography (HPSEC) were used to prepare the multiple fingerprints of 10 batches of polysaccharides from Saposhnikoviae Radix with different production areas and harvest time, and the chemometrics analysis was performed. It presents scientific relevance for the area of Pharmacy, Chemistry, Natural products, Biology and, others area. The language (English) is satisfactory (I suggest the final revision)! However, you need to change some details/information in the title, abstract, Methods, results, discussion and conclusions. I request information on the procedures and interpretation of the results obtained.

Title: Adequate! But, Saposhnikoviae Radix must be written in italics, respecting the genus and species. To review throughout the manuscript, if necessary.

Response: Thanks for your review. The modification has been completed in the full text according to your suggestions.

Abstract: Adequate, but I suggest at the end of the abstract, to highlight the "innovative" proposal of the method.

Response: Thanks for your suggestion. The abstract has been modified as you suggested.

* Introduction section: It is well written, but I suggest:

- Genus and species must be written in italics!

- I suggest inserting a paragraph on the analytical and characterization techniques used.

Response: Thanks for your comment. The modification has been completed in the full text according to your suggestions, a description of the analytical and characterization techniques used is supplemented in line 49-55.

* Results and discussion section

- Pages 2-3, in “2.1.1. PMP-HPLC Fingerprint and Similarity Analysis” section: I suggest expanding the discussions, focusing on the optimization of the analytical parameters of the LC method, as well as the analytical validation.

Response: Thanks for your review. In the process of establing HPLC analysis methods, we referred to the literature, and we adjusted the fluidity ratio, time and phosphate buffer salt concentration in the preliminary experiments. The adjustment of phosphate buffer salt concentration has a great influence on the final liquid phase result, which is discussed in line 97-100.

- Pages 11-12, in “2.4. Anti-allergic Activity of SPs” section: Are there other authors in the literature who have determined these substances (or others in the group) in species for comparison? I suggest expanding the discussions!

Response: Thanks for your suggestion. There are two reports in the literature on the antiallergic activity of parsnip polysaccharide, and we add the relevant discussion.This section of the discussion has been expanded with your suggestions, supplementary discussion is in line 312-317.

- I suggest, at the end of the "results and discussion", to write a paragraph summarizing the findings and their impacts on the research proposal.

Response: Thanks for your suggestion. The corresponding content has been added according to your suggestion, the supplement is in line 350-362.

* Material and Methods section:

- The methodological proposals are coherent, but it needs to improve, especially in the analytical aspects.

- Page 13, in “3.1. Materials and Cell” section: How long were the samples stored until the time of analysis?

Response: Thanks for your review. Samples for the experiment were purchased in batches according to the harvest time, and each batch of samples was extracted as soon as possible after purchase. The obtained polysaccharide samples were stored in the dryer, and all the polysaccharide samples were stored for a maximum of 6 months and a minimum of 1 week before analysis.

- Page 13, in “3.2. Preparation of Polysaccharide Extracts” section: Did you follow any references? If yes, enter!

Response: Thanks for your review. The reference for extraction method has been supplemented according to your suggestion, supplementary references are in line370.

- Page 13, line 357, in “3.3. Characterization of SP and multiple fingerprint profiles” section: To replace “mol/L” by “mol L -1”. To review throughout the manuscript, if necessary.

Response: Thanks for your suggestion. The full text has been revised according to your suggestion.

- Page 14, line 375, in “3.3. Characterization of SP and multiple fingerprint profiles” section: To replace “mL/min” by “mL min -1”. To review throughout the manuscript, if necessary.

Response: Thanks for your comment. The full text has been revised according to your suggestion.

- Page 15, in “3.5. Determination of SPs Anti-allergic Activity” section: Why was the Rat RBL-2H3 lineage selected? About the MTT assay, how long were the cells observed? Did you follow any references? If yes, enter! Were SP cytotoxicity assays performed on healthy cells?

Response: Thank you for your review. Other cells, including rat basophil cells (RBL-2H3), mouse MC tumor cells (P815) and human peripheral blood basophil cells (Ku812), are commonly used in modern studies to replace mast cells for anti-allergy studies. Among them, the RBL-2H3 cell degranulation model has the advantages of low economic cost, high sensitivity and good repeatability of detection method, so we selected RBL-2H3 cells for the experiment of anti-allergic activity. Combining references and pre-experiments, the incubation time we chose after adding MTT is 4 hours, and I have added relevant references in line 428. Since the experimental object was RBL-2H3, SP cytotoxicity test was performed on RBL-2H3 cells.

- Page 15, in “3.5. Determination of SPs Anti-allergic Activity” section: In line 411, to replace “μg/mL” by “μg mL-1”. To review throughout the manuscript, if necessary. In lines 417-437: long paragraph! I suggest splitting the text!

Response: Thank you for your advice. The full text has been revised according to your suggestion.

- Page 15, in “3.6. Spectrum-effect relationship analysis” section: I suggest expanding the information about the correlation analysis, as well as indicating a reference!

Response: Thank you for your advice. Information related to spectrum-effect relationship analysis and references have been supplemented in line 468-471 according to your suggestion.

What are the analytical validation parameters used for methods? Has the proposed method been validated? If so, which protocol / guidelines did you follow? What are the validation parameters studied? Precision, accuracy, LOD, LOQ, robustness, etc. What concentration levels are used to assess accuracy? I suggest detailing the proposed method in more detail...

Response: Thank you for your review. Fingerprint research is qualitative research, and the process of establishing fingerprint follows the "Technical Guiding Principles for Quality Standard Research of New Chinese Medicine (Trial)" issued by the Center for Drug Evaluation, National Medical Products Administration of China. Referring to the previous fingerprint research, the methodological investigation items generally include precision, repeatability and stability. We investigated the precision, repeatability and stability of the fingerprint, and the results were in line with the requirements of Chinese pharmacopoeia.

* Conclusion section

- Adequate! Long paragraph! I suggest splitting the text! Also, I suggest to indicate disadvantages/limitations of the method and the study!

Response: Thank you for your advice. It has been modified and supplemented according to your suggestion in line 502-511.

* Tables and Figures: Adequate!

* Supplementary data: Adequate!

* References: Please, check if the references are in accordance with the journal's rules.

Response: Thank you for your advice. The reference format has been checked.

Reviewer 3 Report

The study presented by the authors is very important, and brings to the attention of specialists, the role of each component in a polysaccharide fraction and scientifically justifies a very important traditional use, the antiallergic effects.

The authors studied a species from the Asian flora, recognized as a traditional remedy in China.

The aim this study is to characterize physico-chemically the polysaccharide fraction and to establish the parameters for determining its quality, correlated with the potential therapeutic effects induced by the components of this fraction.

I consider that study is very important because it can be used as a model in subsequent characterizations of different type of polysaccharide fractions in order to eliminate those components from the carbohydrate chain that may be responsible for inducing certain types of side effects, as the authors point out responsible for the allergic reaction.

The authors use 10 sample of Saposhnikoviae radix, collected from 10 different regions of China, in the study. The method of isolation and purification of the polysaccharide fraction is presented.

The authors use several methods of analysis in determining the chemical profile of the isolated fraction. Modern methods of phytochemical analysis are used, phytochemical profiles and fingerprints are characterized, clusters, hierarchy of main components are analysed, statistical analyses.

 The antiallergic activity is determined, precisely in order to establish the component of the polysaccharide fraction responsible for this potential side effect. It was also established the concentration range at which the polysaccharide fraction exerts antiallergic action, which is a justification for the traditional use of the medicinal species. The authors conclude based on the obtained results, that the presence of glucose and rhamnose in the carbohydrate chain has a very important role in the installation of the antiallergic effect.

The results and discussions are punctual, on each type of analysis performed. Also in this chapter, the authors include numerous diagrams, dendrograms, spectra, graphs, very useful in the punctual evaluation of the research carried out.

The conclusions briefly present the results obtained.

The bibliography supports the research research.

I agree with the publication in the form presented by the authors.

Author Response

Dear Reviewer,

Thank you for your comments on our article entitled " Multiple fingerprints and spectrum-effect relationship of polysaccharides from Saposhnikoviae Radix" (Manuscript ID: molecules-1826140). These comments are very valuable and helpful for us to revise the manuscript. Accordingly, we carefully studied the comments and revised the manuscript. Some of the revised fonts were marked in red. Itemized responses to the comments are listed below the letter. We hope the modification can meet the requirements of reviewers. Looking forward to your reply.

Thank you and best wishes.

Yours sincerely,

Guang Xu.

Your original comments and my response are as follows

The study presented by the authors is very important, and brings to the attention of specialists, the role of each component in a polysaccharide fraction and scientifically justifies a very important traditional use, the antiallergic effects.

The authors studied a species from the Asian flora, recognized as a traditional remedy in China.

The aim this study is to characterize physico-chemically the polysaccharide fraction and to establish the parameters for determining its quality, correlated with the potential therapeutic effects induced by the components of this fraction.

I consider that study is very important because it can be used as a model in subsequent characterizations of different type of polysaccharide fractions in order to eliminate those components from the carbohydrate chain that may be responsible for inducing certain types of side effects, as the authors point out responsible for the allergic reaction.

The authors use 10 sample of Saposhnikoviae radix, collected from 10 different regions of China, in the study. The method of isolation and purification of the polysaccharide fraction is presented.

The authors use several methods of analysis in determining the chemical profile of the isolated fraction. Modern methods of phytochemical analysis are used, phytochemical profiles and fingerprints are characterized, clusters, hierarchy of main components are analysed, statistical analyses.

 The antiallergic activity is determined, precisely in order to establish the component of the polysaccharide fraction responsible for this potential side effect. It was also established the concentration range at which the polysaccharide fraction exerts antiallergic action, which is a justification for the traditional use of the medicinal species. The authors conclude based on the obtained results, that the presence of glucose and rhamnose in the carbohydrate chain has a very important role in the installation of the antiallergic effect.

The results and discussions are punctual, on each type of analysis performed. Also in this chapter, the authors include numerous diagrams, dendrograms, spectra, graphs, very useful in the punctual evaluation of the research carried out.

The conclusions briefly present the results obtained.

The bibliography supports the research research.

I agree with the publication in the form presented by the authors.

Response: Thank you very much for your kind work and consideration on publication of our paper. I would like to express my most sincere thanks, thank you!

Round 2

Reviewer 1 Report

It was pleasing to see on the review that the authors have responded to all of the queries raised in the first review of this work.  However, I am not satisfied with the level of response and some key items have not been addressed.

1) English language grammar

2) SEC data

3) The description of the material source and handling is in my opinion still vague.

I do recognize that this is a large study with a complex data set and significant statistical analysis therein, however, the key points below form a significant part of the reporting and data-set so must be addressed.

For (1), please consider just in the very first sentence of the starting section:

31 ...and is used widely...

It may be wild, and even used wildly, but I would expect some description of such behaviour - are they very spicy dishes, is thrown around the room in a spectacular fashion, is it highly decorative?  I think not.

34.  The us of ...and so on. This is vague and not acceptable.

36.  ...and work has focused on teh structure and related activity of these compounds.

37. The major polysaccharide isolated from Saposhnikoviae Radix has demonstrated a range of bioligical activities [6-9]. However...

These are in individually minor considerations, but overall it leads to a very disjointed report and I must insist that this is corrected.  Throughout the work.

For (2) the authors have not addressed my concerns:

a) The peak shape of the eluting material is not consistent with any SEC profile that I have recorded or can find in the literature.  At an extreme push I could imagine such a peak shape for something eluting at the exclusion limit of the column - but the elution profile would have to run in reverse order. 

Please consider that for this column the exclusion limits are (I calculate) 9.5 and 21.5 minutes with that vlow rate.  So things are starting to elute at the exclusion limit, this makes calculations very difficulty and I would be intrigued to see the standard elution profiles (which was requested in terms of the full working of the Mn/Mw calculations).  

b) The elution times of the SEC standards and the calculations so completed are not reported. as requested

c) The values reported are still a non-sensical number of digits. It was requested that three significant figures could be reported, but not 6-7.

d) The discussion does not refer to Mn & Mw as requested.

For (3) it is excellent that a large sample was used.  But, I still can not find exactly how this was compiled.  For each sample was X-roots sliced from Y locations and these locations were etc.  More precision is needed, it is too vague.  And also, I was very specific that I needed to see if this was done in duplicate or triplicate.  The authors need to spell out that by using such a large sample size they feel this covers off the issue of multiple repeats.  Stating "it was gauranteed" is not scientific, demonstrate this through repeat processes.

Lastly, I am uncomfortable with the lack of yield data.  This is a critical piece of information and should be reported to demonstrate that a) the yield is the same in at least duplicate preparations and b) whether this varies between sites/samples.

Again - I wish to state that I recognise there is a lot of good work completed in this article but the English language use, the unacceptable SEC data and the lack of yield data is a significant stumbling block for this referee.

Author Response

Dear reviewer, my response letter contains pictures, so please check the attachment. Thank you very much.

Reviewer 2 Report

After corrections, I consider the manuscript accepted for publication.

Author Response

Thank you very much for your kind work and consideration on publication of our paper. I would like to express my most sincere thanks, thank you!
